# Location-Aware Range-Error Correction for Improved UWB Localization

**DOI:** 10.3390/s24103203

**Published:** 2024-05-17

**Authors:** Sander Coene, Chenglong Li, Sebastian Kram, Emmeric Tanghe, Wout Joseph, David Plets

**Affiliations:** 1WAVES Group, Department of Information Technology, Ghent University-imec, 9052 Ghent, Belgium; sander.coene@ugent.be (S.C.); emmeric.tanghe@ugent.be (E.T.); wout.joseph@ugent.be (W.J.); 2College of Electronic Science and Technology, National University of Defense Technology, Changsha 410073, China; chenglong.li@nudt.edu.cn; 3Information Technology (Communication Electronics), University Erlangen-Nuernberg, 91058 Erlangen, Germany; sebastian.k.kram@fau.de

**Keywords:** signal processing algorithms, time of arrival estimation, localization, ultra wideband technology

## Abstract

In this paper, we present a novel localization scheme, location-aware ranging correction (LARC), to correct ranging estimates from ultra wideband (UWB) signals. Existing solutions to calculate ranging corrections rely solely on channel information features (e.g., signal energy, maximum amplitude, estimated range). We propose to incorporate a preliminary location estimate into a localization chain, such that location-based features can be calculated as inputs to a range-error prediction model. This way, we can add information to range-only measurements without relying on additional hardware such as an inertial measurement unit (IMU). This improves performance and reduces overfitting behavior. We demonstrate our LARC method using an open-access measurement dataset with distances up to 20 m, using a simple regression model that can run purely on the CPU in real-time. The inclusion of the proposed features for range-error mitigation decreases the ranging error 90th percentile (P90) by 58% to 15 cm (compared to the uncorrected range error), for an unseen trajectory. The 2D localization P90 error is improved by 21% to 18 cm. We show the robustness of our approach by comparing results to a changed environment, where metallic objects have been moved around the room. In this modified environment, we obtain a 56% better P90 ranging performance of 16 cm. The 2D localization P90 error improves as much as for the unchanged environment, by 17% to 18 cm, showing the robustness of our method. This method evolved from the first-ranking solution of the 2021 and 2022 International Conference on Indoor Position and Indoor Navigation (IPIN) Competition.

## 1. Introduction

Location-based services rely on accurate positioning to enable use-cases, from navigation and advertising to emergency positioning and resource allocation. For outdoor scenarios, a Global Navigation Satellite System (GNSS) is the go-to solution; however, for indoor locations that are GNSS denied, other solutions are necessary. In the last few decades, research interest has grown significantly as the cost, size, and prevalence of radio frequency-based technologies have improved. Technologies such as Wi-Fi, Bluetooth low energy (BLE), radio frequency identification (RFID), and ultra wideband (UWB) already enable use cases ranging from automatic control of lights [1] and heating, ventilating, and air conditioning (HVAC) [2], to tracking industrial assets in warehouses [3].

Wi-Fi and BLE, in their most ubiquitous implementations, rely on the signal strength to estimate the distance between sender and receiver, which suffers from shadowing effects and ultimately leads to low localization precision, up to several meter. For certain applications, such as occupancy detection [4], this can be sufficient. In contrast, UWB has the ability to provide fine-grained positioning accuracy due to high ranging resolution, which makes it suitable and promising for the various use cases in the industrial environment [5,6,7]. However, it is not immune to accuracy deterioration. In impulse response (IR) UWB, ranging estimation is performed by finding the first path component (FPC) in the channel impulse response (CIR). Signal propagation in obstructed-Line-of-Sight (OLoS) or non-Line-of-Sight (NLoS) conditions makes this problem harder and can lead to an over- or underestimation of the distance between transmitter and receiver.

Much research has already been done on the mitigation of this ranging error. Many wireless devices support the collection of physical (PHY) layer channel information, which can include the signal phase, channel state information (CSI), or CIR, all of which can be leveraged to estimate the error [8,9,10,11,12]. Some solutions aim to improve localization directly and leverage specular signal reflections on walls. This requires either a priori knowledge of the floor plans [13] or a computationally complex estimation-and-association approach [14]. Others try to classify the propagation conditions using a classifier based on channel features [15], but this is only a partial solution as no ranging or localization improvement can be obtained (or quantified) directly. Another approach is to focus solely on the ranging error and try to model the error based on a set of features, which includes our own work presented here. In [16], the authors expanded the work of [15] and trained a regression model based on the same channel features to predict the ranging error. In [17], the authors propose real-time channel features that are immediately available for commodity UWB devices for NLoS classification and mitigation, in the context of the human shadowing effect. However, this requires two models, one for classification and one for mitigation (in case of NLoS); additionally, they are not able to improve the ranging in Line-of-Sight (LoS) conditions. More recently, deep learning (DL) techniques for NLoS classification and correction have been studied in [18,19,20,21]. In [19], the authors achieve the best results with a weighted least squares (WLS) localization algorithm performed on ranges corrected with a convolutional neural network (CNN) trained on the raw (partial) CIR. This showed an improvement over classical models with handcrafted features, such as in [15,16]. However, they do not provide the network architecture and require concurrent measurements from all involved anchors. A representation learning technique to extract semantic features from the raw CIR and to mitigate the ranging errors is presented in [21], with good range correction results, but was held back by the Gauss–Newton non-linear optimization used for localization in a scenario that requires concurrent range estimations. In [22], the authors propose a generative DL approach and use a variational autoencoder (VAE) to encode a partial CIR. This enables out-of-distribution detection and anomaly score prediction. The anomaly score is then integrated into an extended Kalman filter (EKF) to improve the localization error. Another work investigates CIR-based fingerprinting [23], showing improved localization accuracy in multipath-heavy scenarios, but the approach is limited to one specific environment as the suggested CNN is an end-to-end solution, transforming the input CIR into a 3D coordinate pair.

Whereas the above-mentioned approaches are limited to channel-based information, this paper introduces a novel approach to the range-correction model. We propose the Location-Aware Ranging Correction (LARC) algorithm, in which a preliminary localization loop is used to calculate location-based features. Together with well-known features calculated from channel information, a robust regression model can more precisely predict the ranging error, leading to an improved final localization result.

Figure 1 illustrates the difference between the existing methods and LARC. In uncorrected ranging (Figure 1a), the range estimate is used as is. This leads to non-optimal localization. Existing methods perform a range correction (Figure 1b) to achieve improved localization, but the correction model is limited to features obtained from channel information and/or the range estimate itself. With LARC (Figure 1c), we calculate preliminary location estimates from the uncorrected ranges. Based on the location estimates, we calculate a novel set of location-aware features, such as location, orientation, and velocity. These features, in addition to the more classical channel-based features, are then used by a predictive model to obtain the prediction of the range error. Finally, the corrected range estimations are used to calculate a more accurate final trajectory estimate.

We present an evaluation of our method on an open-access dataset and investigate performance when the environment is modified with respect to the original training environment. In this work, we show the LARC algorithm applied with a Gaussian process regression model (inspired by [16]), using a particle filter (PF) for localization, but any of the components for ranging, localization, or the error prediction model can be exchanged for a different implementation.

The remainder of this paper is organized as follows. Section 2 states the involved UWB-based localization problem. Section 3 presents the details of the proposed UWB ranging and tracking algorithms. We introduce the measurement setup in Section 4 and evaluate and analyze the ranging and tracking performance in Section 5, respectively. Section 6 concludes this paper, with a discussion on the limitations and future work.

## 2. Problem Statement

We provide a general problem statement that describes the scope in which our method can be applied and the assumptions made. The specific instance of this problem that is used to validate our method is discussed in Section 4.

The localization problem as tackled by our approach is as follows. An unknown environment contains a set of static UWB anchors at known locations. We are tasked with locating a nomadic user agent at any given point in time. The only information available is an estimated initial location and a series of CIRs. If the initial location is not given, it can alternatively be calculated based on the range estimates using a least-squares approach. A dataset consisting of CIRs measured over a trajectory with known ground truth locations is available as a training dataset.

Each CIR represents the wireless channel between a tag-anchor pair, at a given time, *t*. The CIR in the complex baseband can be given in general as
(1)$$h\left(\tau ,t\right)=\underset{\mathrm{LoS}}{\underset{︸}{\alpha_{0}\left(t\right)\delta \left(\tau -\tau_{0}\left(t\right)\right)}}+\underset{\mathrm{MPCs}}{\underset{︸}{\sum_{l=1}^{L}\alpha_{l}\left(t\right)\delta \left(\tau -\tau_{l}\left(t\right)\right)}}+n\left(t\right),$$
where $\alpha_{l}$ and $\tau_{l}$ denote the complex amplitude and delay of the *l*-th signal component. l=0 represents the first path component. *L* is the number of MPCs and *n* the measurement noise. Figure 2 shows CIR measurements collected by the commodity UWB device in a warehouse in LoS, OLoS, and NLoS conditions. OLoS is the case where the LoS signal is obstructed by some objects but still detectable with a weaker signal strength. Both OLoS and NLoS propagation between tag and anchor will challenge the accurate range estimation and localization.

For this problem, we assume that the CIRs are measured sequentially, but equitemporal spacing of the measurements is not guaranteed; our solution is also applicable in more lenient cases. Depending on the agent’s location, some anchors might be too obstructed or too far away to provide a CIR measurement (and therefore range estimate). Additionally, we assume causal tracking, i.e., calculating a position estimate at a point in time can only make use of measurement information of the past up to (and including) that point. For real-world applications, this causality is expressed by the term real-time, which introduces a further restriction on the processing time needed to produce a location estimate. For some applications, this restriction might not be present; our approach can also be used in that case.

## 3. Localization Approach

In this section, we explain our approach to tackle the problem described in Section 2. Figure 1 provides a head-to-head comparison between standard and state-of-the-art localization approaches (a, b) and our method (c). The blocks represent the steps in the algorithm flow, each of which are detailed in the sections below. For each CIR input/measurement, the different steps produce outputs with the following notation: *r* is the range estimate, {s} is the set of features, P is the uncorrected location estimate, $r^{\prime }$ is the corrected range estimate, and $P^{\prime }$ is the corrected location estimate. Note that the feature set, {s}, calculated in (c) differs from the one in (b): our proposed method (c) incorporates location information from a preliminary estimation (the first two steps of (c) are in fact approach (a)). The use of the training data for each step is mentioned where applicable.

Our method can be included in different tracking algorithms. We present our results on one specific ranging algorithm and use a particle filter for localization. The choice of the ranging algorithm and localization algorithm is arbitrary, and other options can of course be used as an alternative.

### 3.1. Distance Estimation

Distance estimation is composed of two parts and produces a single distance estimate, given a CIR.

#### 3.1.1. Ranging

As a first step, we apply a ranging algorithm [24] to obtain a distance estimate for each CIR. The ranging algorithm uses a noise-based threshold stemming from a CIR partitioning strategy and identifies the FPC. Local maxima of the CIR magnitude are classified as either noise or signal (plus noise) using linear regression lines in a small area on both sides of each peak. The region from the first CIR bin up to the last noise-only peak produces the noise statistics used to calculate the FPC threshold. The training set is used to optimize the algorithm parameters using a particle swarm optimizer that minimizes the mean absolute error (MAE) of the obtained range estimates with respect to the ground truth ranges. The details of the algorithm and the optimization are presented in [24].

#### 3.1.2. Bias Correction

The on-chip algorithm identifies the rising slope of a pulse, while our approach identifies the pulse peak. This introduces a small time offset. A bias correction term is therefore subtracted from the range estimate to account for antenna delay and bin timing interpretation. The median ranging error (using the optimal ranging algorithm parameters) is taken as range bias. It is calculated per anchor using the complete training dataset.

### 3.2. Particle Filter Tracking

We use a 2D PF [25] to obtain the location updates of the moving tag. A collection of $N_{\mathrm{particles}}$ particles represents possible solution states of the nomadic user, at a single point in time (k). One such particle/state, $S_{i}^{\left(k\right)}={\left[X^{\left(k\right)},Y^{\left(k\right)},v_{X}^{\left(k\right)},v_{Y}^{\left(k\right)}\right]}_{i}^{T}$, contains information on the position, $P_{i}^{\left(k\right)}={\left[X^{\left(k\right)},Y^{\left(k\right)}\right]}_{i}^{T}$, and velocity, $v_{i}^{\left(k\right)}={\left[v_{X}^{\left(k\right)},v_{Y}^{\left(k\right)}\right]}_{i}^{T}$, of the user, with $i=1\ldots N_{\mathrm{particles}}$ being the index of the particle; additionally, each particle carries a certain weight, $w_{i}^{\left(k\right)}$, with $\cdot^{T}$ being the transpose operator. *X* and *Y* are geometrical coordinates for a chosen coordinate system (in which the anchors are localized); $v_{X}$ and $v_{Y}$ are their respective velocity components. When the variables are used with $\cdot_{\left(i\right)}$, we talk about a specific particle; without the subscript, we mean the global variable. A global location estimate is obtained with a weighted mean of all particles:(2)$$S^{\left(k\right)}=\sum_{i=1}^{N_{\mathrm{particles}}}w_{i}^{\left(k\right)}S_{i}^{\left(k\right)}.$$

As the problem statement provides an initial position estimate, the particle filter is initialized with equal-weight particles distributed following a 2D normal distribution with standard deviation of 1 m around the initial estimate. If this is not available, alternative initialization can use a uniform distribution across the tracking area or by using a trilateration position estimate (starting the PF only after a few measurements).

The filtering process is executed in three steps that are executed with each CIR measurement:

#### 3.2.1. Prediction

In the prediction step, the particles are updated via a constant velocity motion model (subscript *i* is dropped for clarity), given by
(3)$$\left\{\begin{array}{cc} X^{\left(k+1\right)} & =X^{\left(k\right)}+v_{X}^{\left(k\right)}\cdot \Delta T^{\left(k\right)}+0.5\cdot {\left(\Delta T^{\left(k\right)}\right)}^{2}\cdot n_{a}\\ Y^{\left(k+1\right)} & =Y^{\left(k\right)}+v_{Y}^{\left(k\right)}\cdot \Delta T^{\left(k\right)}+0.5\cdot {\left(\Delta T^{\left(k\right)}\right)}^{2}\cdot n_{a}\\ v_{X}^{\left(k+1\right)} & =v_{X}^{\left(k\right)}+\Delta T^{\left(k\right)}\cdot n_{a}\\ v_{Y}^{\left(k+1\right)} & =v_{Y}^{\left(k\right)}+\Delta T^{\left(k\right)}\cdot n_{a} \end{array},\right.$$
where $\Delta T^{\left(k\right)}=T^{\left(k+1\right)}-T^{\left(k\right)}$ denotes the elapsed time between the new state at step k+1 and the previous state at step *k*. $n_{a}$ is the process noise, which is in this case an acceleration error. It is sampled from the normal distribution, $n_{a}∼N\left(0,\sigma_{a}\right)$, with zero mean and standard deviation, $\sigma_{a}$. The value of $\sigma_{a}$ is estimated from the training data, which we indicate with ${\hat{\sigma }}_{a}$. Since the ground truth does not include a velocity measurement, the scalar velocity component, $v_{X/Y}^{\left(k\right)}$, is estimated via the consecutive location changes within a given time slot. The acceleration, $a_{X/Y}^{\left(k\right)}$, is calculated using a similar approach. This is performed for both *X* and *Y* separately. We estimate the overall acceleration variance, ${\hat{\sigma }}_{a}^{2}$, using
(4)$${\hat{\sigma }}_{a}^{2}=\frac{1}{2(N-2)-1}\sum_{k=3}^{N}\left[{\left(a_{X}^{\left(k\right)}-{\hat{\mu }}_{a}\right)}^{2}+{\left(a_{Y}^{\left(k\right)}-{\hat{\mu }}_{a}\right)}^{2}\right],$$
where N−2 is the number of measurements for which the acceleration is calculated. ${\hat{\mu }}_{a}$ denotes the mean scalar acceleration, calculated over both *X* and *Y* components together.

The prediction step is used at each CIR measurement to match up the filter time with the observation time.

#### 3.2.2. Correction

Each measurement observation, $y^{\left(k+1\right)}$, is used to update the weights of all particles,
(5)$${\bar{w}}_{i}^{\left(k+1\right)}=w_{i}^{\left(k\right)}p\left(y^{\left(k+1\right)}|S_{i}^{\left(k\right)}\right),$$
where *p* is the likelihood function, a metric for the probability of the observation conditioned on the current state of the particle. The $\bar{w}$ indicates that the weight is not normalized. After performing the weight calculation for all particles, the weights are normalized:(6)$$w_{i}^{\left(k+1\right)}=\frac{{\bar{w}}_{i}^{\left(k+1\right)}}{\sum_{j=1}^{N_{\mathrm{particles}}}\;{\bar{w}}_{j}^{\left(k+1\right)}}.$$

The observation $y^{\left(k+1\right)}$ contains the anchor position, $P_{\mathrm{anch}}$, and the range, *r*, estimated from the CIR. For each particle, the distance, $r_{i}$, is calculated and compared to the CIR-based range estimate through a probability density function (PDF), *f*, of the ranging error, such that
(7)$$p\left(y|S_{i}\right)=f\left(r-\underset{r_{i}}{\underset{︸}{\left|\right|P_{i}-P_{\mathrm{anch}}{\left|\right|}_{2}}}\right),$$
where $P_{i}$ is contained within the state vector, $S_{i}$, and *r* and $P_{\mathrm{anch}}$ are contained within the observation vector, y; ${\left||\cdot |\right|}_{2}$ is the Euclidean norm.

The training set is used to determine the PDF, through a fitting process. Note that we obtain two such functions: one fitted to the original, uncorrected ranging errors ($f_{\mathrm{uc}}$), and one fitted to the corrected ranging errors ($f_{c}$). The error prediction model can overfit on the training dataset, leading to an optimistically low error spread. We therefore opt for a parametric error distribution, as this enables us to create a new averaged distribution, $f_{\mathrm{av}}$, with parameters averaged from the parameter values of $f_{\mathrm{uc}}$ and $f_{c}$. $f_{\mathrm{uc}}$ is used unmodified in the first PF; $f_{\mathrm{av}}$ is used in the second PF.

The choice of parametric distribution to fit the error distribution is influenced by the used ranging algorithm and propagation conditions. The error distribution is not normally distributed; the t-location scale distribution achieves a much better goodness-of-fit. The PDF of the t-location scale distribution family is given as follows,
(8)$$f\left(x;\mu ,\sigma ,\nu \right)=\frac{\Gamma (\frac{\nu +1}{2})}{\sigma \sqrt{\nu \pi }\Gamma \left(\frac{\nu }{2}\right)}{\left[\frac{\nu +{\left(\frac{x-\mu }{\sigma }\right)}^{2}}{\nu }\right]}^{-\frac{\nu +1}{2}},$$
where Γ(·) is the gamma function. μ denotes the location parameter, σ the scale parameter, and ν the shape parameter.

Evaluation of the analytical expression for the PDF is computationally expensive because it is executed for each particle’s range observation individually. Therefore, at initialization of the PF, we evaluate the PDF at a number of (1D) grid points. The analytically evaluated points are stored, creating a look-up table, and used for linear interpolation during execution of the PF. A look-up table directly sampled from the empirical error distribution is not used to avoid overfitting. As $\lim_{x\to \pm \infty }f\left(x\right)=0$, only a limited central portion of the PDF needs to be evaluated. Therefore, we chose the grid to cover the central 95% of the PDF, sampled uniformly at 1 cm intervals. Points outside the grid are set to their nearest boundary value. The PDF itself is fit on the complete empirical error distribution.

#### 3.2.3. Resampling

We resample the particle set after each correction (i.e., bootstrap particle filter). The new generation of particles is generated using a Monte Carlo method. They are initialized as copies of the existing particles, using their normalized weights as creation probability.

### 3.3. Range-Error Prediction Model

To achieve optimal positioning, we reduce the ranging error by predicting the expected error based on a set of inputs, i.e., features, {s}. Section 1 described the common approach of features that rely on channel information alone. Here we augment the number of features using the preliminary tracking results (position, P, velocity, v, and derived metrics).

The chosen model is a Gaussian process regression model [26] with an exponential kernel and a linear basis function.

#### 3.3.1. Features

Table 1 lists the features used in our model. Features that use previous measurement points make use of the (k) indexing. Figure 3 provides a visual companion to the trajectory features presented below. For each CIR measurement, these features are obtained as follows:

anchor: represents the signal origin as an integer identifier. Its numerical value is categorical and of no importance.*r*: the estimated (debiased) range from the ranging algorithm for the current measurement.logFPSS: we calculate the first path signal strength (FPSS) as a pseudo received signal strength indicator (RSSI), heuristically approximated around its FPC as
(9)$$\mathrm{FPSS}=|\mathrm{CIR}\left(i_{\mathrm{FPC}}\right){|}^{2}+{\left|\mathrm{CIR}\left(i_{\mathrm{FPC}}+1\right)\right|}^{2},$$
with $i_{\mathrm{FPC}}$ the (rounded down integer) bin index of the FPC. The ranging algorithm subsamples; the enclosing integer bin indices are therefore $i_{\mathrm{FPC}}$ and $i_{\mathrm{FPC}}+1$.logmax|CIR|: measure for the peak magnitude of the CIR, calculated as
(10)$$\log\max|\mathrm{CIR}|=\log\sum_{i=-1}^{1}{\left|\mathrm{CIR}\left(i_{\max}+i\right)\right|}^{2},$$
with $i_{\max}$ being the bin index of the maximum magnitude of the CIR.X,Y: the position coordinates obtained with the particle filter.ω: the heading of the user, in radians, calculated with
(11)$$\omega \left(k\right)=\left\{\begin{array}{cc} 0 & k=1,\\ \tan^{-1}\left(\frac{Y^{\left(k\right)}-Y^{\left(k-1\right)}}{X^{\left(k\right)}-X^{\left(k-1\right)}}\right) & k=2,\\ \tan^{-1}\left(\frac{3Y^{\left(k\right)}-4Y^{\left(k-1\right)}+2Y^{\left(k-2\right)}}{3X^{\left(k\right)}-4X^{\left(k-1\right)}+2X^{\left(k-2\right)}}\right) & k\geq 3, \end{array}\right.$$
with *k* being the measurement index of the current measurement. The discrete approximation, for k≥3, reduces noise in position estimates.ϕ: the angle-of-departure from the anchor, in radians, using world coordinates, calculated using
(12)$$\phi =\tan^{-1}\left(\frac{Y-Y_{\mathrm{anch}}}{X-X_{\mathrm{anch}}}\right),$$
with $\left(X_{\mathrm{anch}},Y_{\mathrm{anch}}\right)$ being the coordinates of the communication’s anchor, at the measurement time.θ: the angle-of-arrival with respect to the user, in radians, using local coordinates, calculated using
(13)θ=ϕ−ω+π.
While this is mathematically redundant information, the model still performs better when it is a separate feature. This is likely due to the model kernel.*v*: the total scalar velocity of the user, calculated using
(14)$$v\left(k\right)=\frac{∥P^{\left(k\right)}-P^{\left(k-1\right)}∥}{\Delta T^{\left(k\right)}}.$$ξ: the difference in range between the position estimate and the original estimate from the ranging algorithm. It is calculated using
(15)$$\xi =r-∥P_{\mathrm{anch}}-P∥,$$
with $P_{\mathrm{anch}}$ being the location of the anchor and P the current position estimate. This is inspired by the fact that a perfect position estimate yields the ranging error for the feature.

The first four features are available from the ranging algorithm and the measurement information. The others rely on estimated positions, which is the novelty of our approach.

#### 3.3.2. Training

The training dataset uses the aforementioned features in combination with the ranging error calculated from the ground truth to train the regression model. We distinguish two types of ranging errors:Precision errors: These errors stem from the limited bin timestamp precision as well as noise when the FPC is attenuated.Accuracy errors: These errors stem from wrong identification of the FPC. This can happen when (i) the calculated ranging algorithm threshold is inadequate for the CIR, (ii) the FPC is not contained in the captured CIR, or (iii) the FPC is indistinguishable from noise due to too much attenuation. Accuracy errors are corrected for in the localization steps: deviant measurements are filtered out due to low particle weights.
It is the former kind we target with the regression model. To avoid inclusion of the latter kind for training, we filter measurements based on the following selection criteria
FPC bin index: The accumulator auto-centers the estimated FPC of a CIR on a specific index. The centering is based on the initial FPC identification. When the FPC identified by our ranging algorithm is closer than 4 ns to this center, we know our estimation is reliable, and it is included in the model training. This can be adjusted for the time resolution of the CIR samples and the expected accuracy of the ranging algorithm. A FPC identified earlier indicates enough attenuation for the FPC to be missed at the initial centering, an indicator of NLoS measurements.Radial velocity: We calculate the radial velocity with respect to each anchor, based on the range estimates. A velocity $|v_{r}|>5\;m/s$ indicates erroneous range estimates due to unrealistically high speed (for the problem of human and slow robotic motion, this can be adjusted to fit the problem). We exclude the measurements with overly high velocity, as well as the two measurements following it.
Note that both these criteria do not rely on the location estimate. During application of the localization pipeline, outlier ranges are removed with the same method and excluded from the final particle filter pass. Training on the complete set results in slightly better ranging correction, but slightly worse positioning performance.

## 4. Measurement Setup

The International Conference on Indoor Navigation and Indoor Positioning (IPIN) has been hosting a competition to further research on the topic of indoor localization and tracking [27,28,29]. Track 7 of the competition focuses on the problem of cooperative UWB positioning and tracking in challenging environments using commodity devices [30]. It provides training and competition CIR data to enable a direct head-to-head comparison of contestants’ approaches. The competition data and the corresponding technical annex are open-access [30] and facilitate further investigation from the community. Using the competition data, we investigate the use of location-aware features.

Our proposed method was conceived at the IPIN Competition 2021, which allowed for non-causal, offline processing of the CIR measurements, i.e., a smoothing approach. Our first-place results in the competition [29,31] are therefore slightly better than the ones we present in this paper, where we self-impose the more realistic restriction of causal measurement processing.

### 4.1. Environment

The environment consists of an area of approx 300 m^2^ in an industrial warehouse. Reflecting and absorbing elements are present, such as the walls of the measurement hall, metal gates and artificially included reflector/absorber wall elements, industrial vehicles, and metal shelves. The left diagram of Figure 4 schematically sketches the environment, while the real-world environment is shown on the right. The receiving anchors are placed around the recording area at approximately 1.5 m height. The transmitter device is carried by a human/worker at chest height and regularly transmits UWB signals received by the anchors.

### 4.2. Equipment

The ground truth of the transmitter positions is collected using a millimeter-accurate Qualisys motion tracking system. The data is collected and synchronized by a network time protocol (NTP) server and pre-processed (corrupted data points are removed and radio frequency (RF) and positioning reference data are synchronized).

The data are recorded using a platform based on the Decawave DW1000 UWB chip, which provides nanosecond resolution in the delay domain. The device is configured to use channel 2 (4 GHz center frequency and 499.2 MHz bandwidth), with an omnidirectional antenna.

### 4.3. Datasets

In total, there are three datasets: the training set, the test 1 set, and the test 2 set. All sets have a measurement sampling rate of 26 Hz. The training set contains 28,000 measurements, captured over 1074 s. Both test sets contain 5000 measurements, captured over 193 s. Figure 5 shows the trajectories that correspond to each dataset.

The training set and test set 1 are recorded in the exact same environment. Test set 2 presents a modified scenario; in this setup, clutter elements are moved around within the environment (e.g., forklift, van) which leads to a different propagation scenario. Test set 2 is therefore ideal to evaluate how well an error-mitigation approach can be generalized and how it adapts to changes in the environment.

## 5. Results

In this section, we apply the pipeline to the open-access data discussed earlier. We quantify the performance of the ranging correction and final 2D trajectory using the cumulative distribution function (CDF) of the error, as well as scalar metrics such as the MAE, the root mean square error (RMSE), and percentile values (e.g., P90 for the 90th percentile).

To illustrate the performance of our method, we compare three approaches, following the notation in Section 3 and Figure 1:(a)An uncorrected approach.(b)A correction that uses features based on channel information; we follow [15], using the signal energy, maximum amplitude, mean excess delay, root mean square delay spread, kurtosis, and estimated range. Rise time cannot be included because of the lower time resolution of the CIRs. We replaced the least squares (LS) support vector machine (SVM) with the same exponential Gaussian process regression as in our proposed method. The LS-SVM worsened the ranging error in all datasets.(c)Our method LARC, a correction that uses features based on both channel information and a preliminary location estimate.
All approaches use the same particle filter implementation for localization, with $N_{\mathrm{particles}}=5000$, using the same resampling strategy and weighing approach. Only the likelihood functions differs as the error PDF, *f*, is adapted to each method:(a)The error PDF is fitted to the ranging error of the training data ($f=f_{\mathrm{uc}}$).(b)The error PDF is fitted to the corrected ranging error of the training dataset ($f=f_{c}$) corrected using method (b).(c)Our method contains two localization passes: PF1 and PF2. PF1 uses the original ranging error PDF ($f_{1}=f_{\mathrm{uc}}$). PF2 uses the parametrically averaged error PDF ($f_{2}=f_{\mathrm{av}}$).
Since the error correction model can overfit on the training set, we construct the PDF for the likelihood of the second localization pass as follows: each t-location scale distribution parameter is averaged between the original PDF (of the uncorrected ranging error), $\cdot_{\mathrm{uc}}$, and the corrected error PDF parameters, $\cdot_{c}$, such that
(16)$$f_{2}=f_{\mathrm{av}}=f\left(x;\frac{\mu_{\mathrm{uc}}+\mu_{c}}{2},\frac{\sigma_{\mathrm{uc}}+\sigma_{c}}{2},\frac{\nu_{\mathrm{uc}}+\nu_{c}}{2}\right).$$

Table 2 and Table 3 summarize the evaluation results using scalar metrics for the ranging and positioning error, respectively. A regular correction method that uses features solely based on channel information (column (b)) overfits on the training set, resulting in worse ranging and positioning performance than compared to our LARC method (column (c)). While we acknowledge the difference in problem statement and measuring equipment that can be at the root of this, it is still an indication that such an approach is not generally applicable nor robust enough.

Execution time of our algorithm is measured using the MATLAB timeit function. The whole localization chain to obtain a final position estimate (i.e., ranging, PF1, feature calculation, correction, PF2) takes 3.4 milliseconds per measurement on an i7-7700 CPU running at a clock speed of 3.6 GHz; we do not make use of multi-threading, graphics processing unit (GPU) acceleration, nor neural processing unit (NPU) acceleration. In other words, our localization algorithm can run in real-time up to an acquisition rate of 294 Hz. The mean acquisition rate of the dataset used here is 25 Hz, meaning our approach can perform at nearly a 12× real-time rate. The GP regression model takes up 3.2 MB of memory.

### 5.1. Training Set

The results of the training dataset are only useful as an indication that error correction can take place. Table 2 lists the numerical results, showing an improvement in ranging precision. Method (b) reduces the P90 error to 3 cm, whereas our LARC method (c) achieves 9 cm, down from 36 cm for (a). Note that, for (c), approximately 27% of the training data are unseen, because of the filtering that happens prior to training the model (see Section 3.3.2). This explains the gap in performance between (b) and (c), which would otherwise seem to indicate a poor model choice. We compare to the results of the training set to identify a potential overfit of the model.

### 5.2. Test Set 1: Same Environment

For the same environment as the training set, i.e., the first test set, Figure 6a shows the ranging error CDF for an unseen trajectory. The method based only on channel information (b) is unable to provide consistent improvement to the ranging, if at all. This indicates that indeed, the method overfits on the training data, resulting in worse performance on unseen data than on training data. The selected features of method (b) are possibly too correlated, leading to an over-dimensioned model. Method (b) mostly impacts outliers, as seen in Table 2, decreasing the ranging error MAE from 24 cm to 19 cm and the RMSE from 79 cm to 35 cm. P90 error is unchanged at 36 cm. These errors are an order of magnitude larger than those on the training set, indicating overfit. Our LARC method (c) is more consistent with training, providing a visible improvement to the ranging error. The MAE decreases by 41% to 15 cm, and the RMSE by 12% to 70 cm, indicating the continued presence of outliers. This is acceptable; Section 3.3.2 explains that our model focuses on the smaller, correctable precision errors. The localization solution manages the impact of outliers; large (outlier) ranging errors do not influence the PF as the weight of all particles will be zero when evaluating the likelihood function. The P90 error decreases by 58%, to 15 cm.

Figure 6b shows the 2D positioning error CDF for the first test set. Interestingly, solution (b) now performs worse, with the MAE increasing from 13 cm to 16 cm, and the P90 error increasing from 22 cm to 28 cm, as listed in Table 3. A possible explanation for this lies in the localization approach. The ranging outliers (P95, P99) are now less pronounced and will therefore influence the trajectory calculations more heavily, resulting in higher 2D errors. Our LARC method (c) does create an improvement for the final trajectory, though less pronounced than for the ranging error. The MAE decreases by 30% to 9 cm. The P90 error decreases by 4 cm to 18 cm (−21%).

### 5.3. Test Set 2: Varied Environment

By varied environment, we mean unseen data with changed locations of some clutter object with respect to the environment of the training set, while keeping the location of the anchor nodes identical. Figure 7a shows the ranging error CDF for the second test set. The overfit of method (b) is clear, as the correction even increases the error. As listed in Table 2, the ranging MAE increases from 30 cm to 32 cm, the P90 error doubles from 37 cm to 74 cm, while the RMSE decreases from 1 m to 69 cm. Indicating again that, while outliers are improved, the majority of localizations deteriorate. Our LARC method (c) shows an important improvement: decreasing the MAE by 30% to 21 cm and the RMSE decreases by 10% to 92 cm. The P90 decreases by 57%, to 16 cm. This is completely in line with the results of test set 1, indicating great robustness to changes in environment.

Figure 7b shows the 2D position error CDF for the second test set. Method (b) fails to improve the trajectory. As explained for test set 1, this is caused by the (slightly) improved outliers that now more strongly influence the localization algorithm, while the bulk of ranging estimates is worse. Our method (c) obtains an improved location estimate, decreasing the MAE by 20%, from 13 cm to 10 cm, as listed in Table 3. The P90 error improvement is equal to test set 1, decreasing by 4 cm to 18 cm (−17%). This once more shows the robustness of our method.

Overall, these results shows that our method can be used to improve UWB ranging and localization using COTS devices.

### 5.4. Feature Ranking

The feature list we present in Section 3.3.1 is extensive. Therefore, we now take the time to analyze the ranking of the features using F-tests, to investigate which features are most important to the model. An F-test is performed on each feature individually and test the hypothesis that response values grouped by predictor variable values are drawn from populations with the same mean against the alternative hypothesis that the population means are not all the same [32].

Figure 8 shows the result of the F-test, ranking the features from high to low importance. The importance score is calculated as −logp, where the *p*-value is the result of the hypothesis test. The highest importance scores are represented as infinity, due to numerical accuracy when the *p*-value approaches machine precision. The most important features are the first path signal strength and ξ, the difference between the user position estimate’s range and the range estimated from the CIR. The least important features are the location coordinates *X* and *Y*, the AOD ϕ, and the anchor identifier. This can have a significant impact for applications and training: *X*, *Y*, ϕ, and the anchor identifier are location-specific features. In other words, these features capture effects from the absolute lay-out of the training environment and are intuitively not suited if the model is deployed at a different location. However, because we have shown that these have the least importance, and no impact on performance, the chosen regression model is portable to other environments when we drop these features from the model.

Figure 9 shows the impact of dropping the features with the lowest importance score. Dropping the four least-important features, such that only seven remain, has no impact on final localization performance. Keeping only the three most important features has a significant impact, increasing the error by 15–20%. The implication is that the (additional) location-based features that were introduced in this paper with a feature importance score of only 65–92 have a meaningful impact and showcase the benefit of using location-based features to achieve good ranging correction.

## 6. Conclusions and Future Work

This paper has introduced the concept of location-aware features to improve ranging estimation errors. The ranging improvement of our method is large (approximately 57% P90 error improvement, to 15 cm) and is consistent across the unseen test datasets, even when introducing changes in the environment. The localization also improves (4 cm improvement to the P90 error, to approximately 18 cm), though it is less pronounced than the ranging improvement.

The use of trajectory-based features such as heading and velocity has a drawback, as it makes training a model using static measurements impossible. For some real-world applications, it might be impractical to hold a measurement campaign to obtain the required training data. A device capable of angle-of-arrival (AoA) estimation could alleviate this drawback, because features such as the user heading can be calculated directly, without the need for trajectory estimation, and thus, with a modified error estimation model (dropping, e.g., velocity features), static measurements could suffice.

Since the features are based on absolute environment location (e.g., *X*, *Y*) as well as relative position and orientation with respect to the anchors (e.g., *r*, θ), it is clear that the resulting error prediction model does not lend itself to re-use in a completely different environment. However, we have shown that it is robust enough to perturbations in the environment, and that the location-specific features can be left out without loss in performance. This allows the model and method to be deployed in dynamic environments, with changing numbers or locations of scatterers. Additionally, in environments like office buildings, where the general layout of each floor is similar, this method might also be applied. This robustness is an improvement over a model based on only channel information, which fails to generalize well to changes in the environment.

In the future, we will investigate the impact of the choice of ranging and localization algorithms on the obtained improvements, as well as the type of prediction model (e.g., CNN).

## Figures and Tables

**Figure 1 sensors-24-03203-f001:**
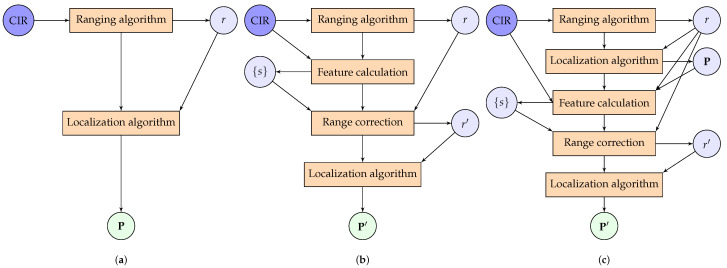
Comparison between localization solutions in the literature and ours: (**a**) is the default approach without any corrections, (**b**) makes use of an error prediction model based on channel information, in (**c**), our proposed LARC solution calculates location-aware features from an initial trajectory estimation. *r* is the range estimate, {s} is the set of features, P is uncorrected location estimate, $r^{\prime }$ is the corrected range estimate, and $P^{\prime }$ is the corrected location estimate. The second localization algorithm uses the same code and parameters, differing only in likelihood function.

**Figure 2 sensors-24-03203-f002:**
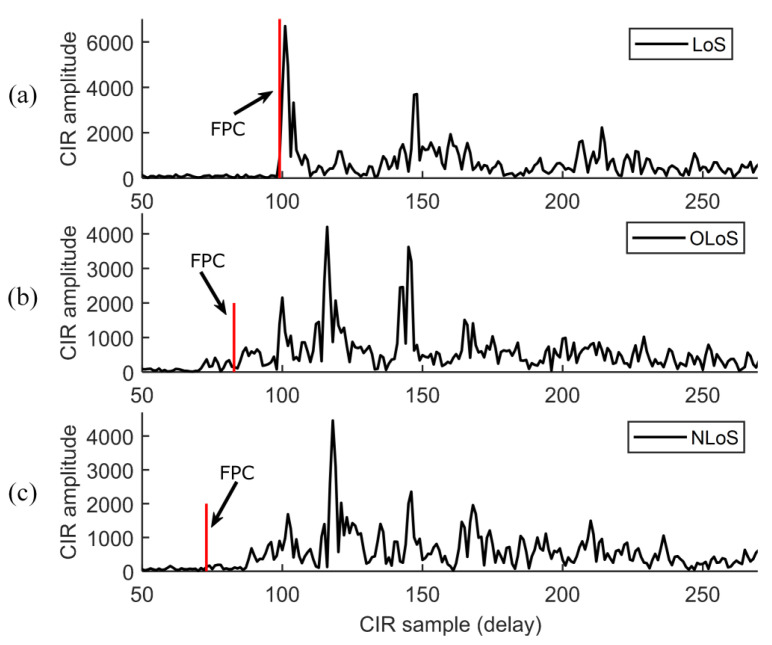
Measured CIR data of the commodity UWB device in the cases of (**a**) LoS, (**b**) OLoS, and (**c**) NLoS propagation. (FPC: first-path component).

**Figure 3 sensors-24-03203-f003:**
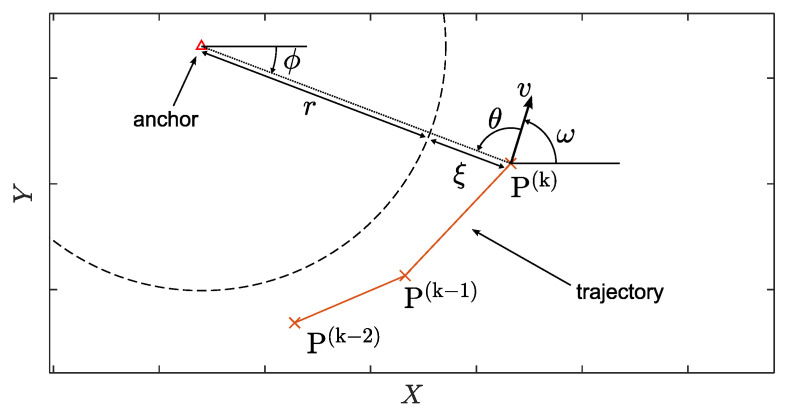
A visual representation of the trajectory-based features.

**Figure 4 sensors-24-03203-f004:**
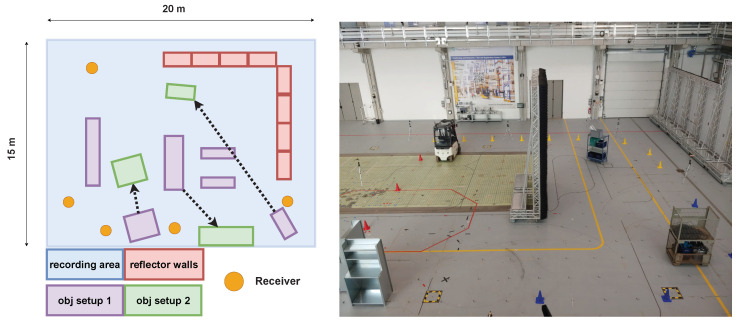
Schematic environment setup, including exemplary object setups 1 and 2 (**left**) and a similar real-world environment (**right**).

**Figure 5 sensors-24-03203-f005:**
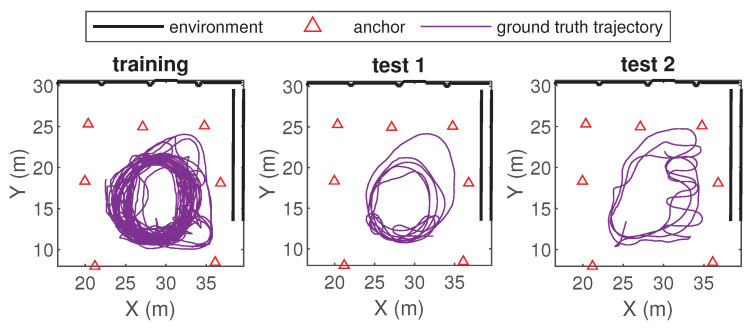
Overview of the UWB deployments and the dataset trajectories. Clutter objects are present in all cases, but are not represented here.

**Figure 6 sensors-24-03203-f006:**
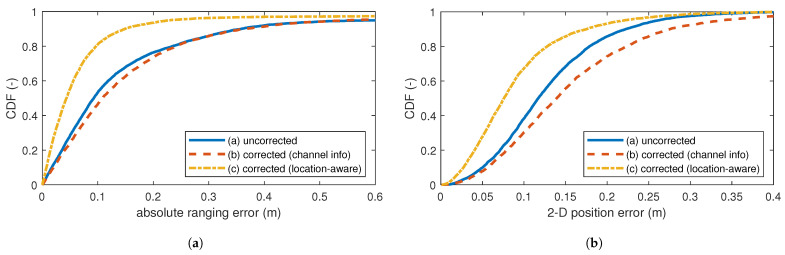
Results of test set 1: CDF of (**a**) the absolute ranging error and (**b**) the 2D position error in meters. Evaluated without and with corrections.

**Figure 7 sensors-24-03203-f007:**
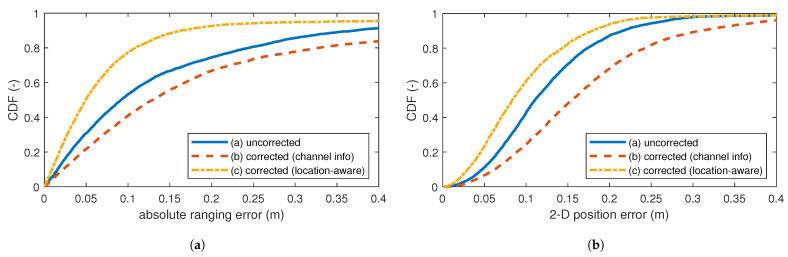
Results of test set 2: CDF of (**a**) the absolute ranging error and (**b**) the 2D position error in meters. Evaluated without and with corrections.

**Figure 8 sensors-24-03203-f008:**
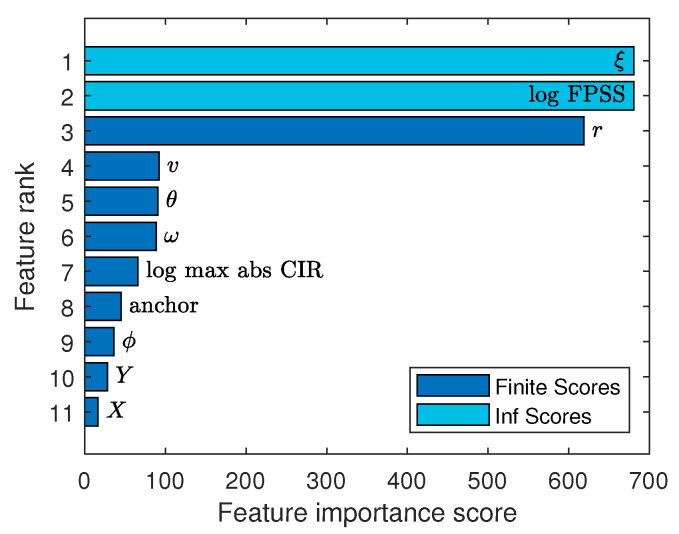
Feature ranking based on F-tests. The importance score is calculated as −logp; a high value indicates that the feature is important.

**Figure 9 sensors-24-03203-f009:**
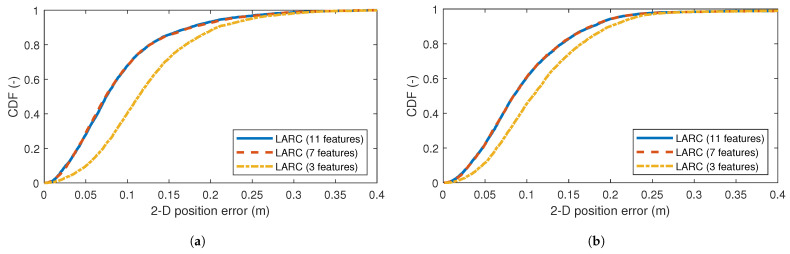
Results of feature selection in our method. CDFs of the 2D position error in meter show the impact of dropping the lowest ranked features. (**a**) shows the results for test set 1 and (**b**) shows the results for test set 2.

**Table 1 sensors-24-03203-t001:** Features used in the regression model.

Predictor	Description
anchor	identifier of signal origin
*r*	estimated range
logFPSS	first path signal strength
logmax|CIR|	CIR peak magnitude
*X*	position estimate coordinate
*Y*	position estimate coordinate
θ	angle-of-arrival to user, in local coordinates
ϕ	angle-of-departure from anchor, in world coordinates
*v*	total scalar velocity of the user
ω	heading of the user
ξ	difference between user position estimate’s range and observed range estimate from CIR

**Table 2 sensors-24-03203-t002:** Resulting absolute ranging errors in meters.

Metric	Training						Test 1						Test 2				
	(a)	(b)/[%]		(c)/[%]			(a)	(b)/[%]		(c)/[%]			(a)	(b)/[%]		(c)/[%]	
MAE	0.26	0.01	−95	0.14	−48		0.24	0.19	−24	0.15	−41		0.30	0.32	+7	0.21	−30
RMSE	0.97	0.02	−98	0.89	−8		0.79	0.35	−56	0.70	−12		1.02	0.69	−32	0.92	−9
P50	0.09	0.01	−87	0.02	−78		0.09	0.11	+16	0.04	−53		0.09	0.13	+41	0.05	−46
P75	0.19	0.02	−89	0.04	−78		0.19	0.21	+9	0.08	−56		0.20	0.26	+29	0.09	−54
P90	0.36	0.03	−92	0.09	−75		0.36	0.36	+2	0.15	−58		0.37	0.74	+98	0.16	−56
P95	0.67	0.03	−95	0.19	−71		0.61	0.55	−10	0.23	−62		0.72	1.55	+114	0.33	−54
P99	4.06	0.04	−99	3.48	−14		4.03	1.57	−61	3.45	−14		5.42	3.26	−40	4.94	−9

(a) without correction, (b) correction using channel information features, (c) LARC: correction using channel information and location-aware features.

**Table 3 sensors-24-03203-t003:** Resulting 2D positioning errors in meters.

Metric	Training						Test 1						Test 2				
	(a)	(b)/[%]		(c)/[%]			(a)	(b)/[%]		(c)/[%]			(a)	(b)/[%]		(c)/[%]	
MAE	0.12	0.02	−81	0.05	−59		0.13	0.16	+22	0.09	−30		0.13	0.18	+39	0.10	−20
RMSE	0.14	0.03	−80	0.06	−56		0.15	0.19	+26	0.11	−23		0.16	0.21	+37	0.14	−11
P50	0.12	0.02	−82	0.04	−62		0.12	0.14	+19	0.08	−35		0.11	0.15	+40	0.08	−25
P75	0.16	0.03	−81	0.07	−58		0.17	0.20	+22	0.11	−31		0.16	0.22	+38	0.13	−20
P90	0.21	0.04	−81	0.10	−54		0.22	0.28	+24	0.18	−21		0.22	0.31	+44	0.18	−17
P95	0.25	0.05	−80	0.12	−52		0.26	0.34	+30	0.22	−16		0.26	0.38	+46	0.21	−19
P99	0.33	0.07	−79	0.17	−47		0.35	0.47	+35	0.32	−7		0.42	0.68	+60	0.44	+4

(a) without correction, (b) correction using channel information features, (c) LARC: correction using channel information and location-aware features.

## Data Availability

No new data were created or analyzed in this study. The existing data used in the manuscript are available at https://owncloud.fraunhofer.de/index.php/s/RURgoPDou3PgF3U (accessed on 10 April 2024) with supporting documentation at https://evaal.aaloa.org/images/2021/IPIN_Track7_v3.pdf (accessed on 10 April 2024).

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
