# Peer review of "Location-Aware Range-Error Correction for Improved UWB Localization"

_sensors, 2024, doi:10.3390/s24103203_

Round 1
Reviewer 1 Report
Comments and Suggestions for Authors
This article proposes a a location-aware ranging correction on ultra-wideband signals. By incorporating a preliminary location estimate into a localization chain, motion-derived features can be calculated as inputs to a range error prediction model. A Gaussian process regression model is proposed that uses two kinds of features including location-aware features and channel information features. Experimental results show the accuracy and robustness when the training set and validation set differ in scenarios of the proposed method.
1, Authors mention in line 35 page 2 that other methods are limited to channel-based information and therefore location-aware features are introduced in this proposed method. The features used is listed in Table 1. What are the contributions of each features in the result? Without relevant analyses in experimental part, it could be difficult to prove the effectiveness of the proposed location-aware features.
2, In line 61, page 2, authors mention that deep learning based ranging error estimation methods have evolved greatly these years. How about compare your method with some SOTA deep learning based methods, which could further prove the robustness of your method in varied environments. Besides, many SOTA methods involving UWB localization is proposed these year. A comparison with these methods proposed in last 2 years may help prove the novelty of your method.
3, In the experimental result part, the corrected method using channel info is even worse than uncorrected result as shown in figure 6 and figure 7, which is counter-intuitive. Please explain it.
4, The abstract of this article mainly focus on the experimental results of your proposed method. It would be better to illustrate in the abstract part how your method out-stands other SOTA method in signal processing or deep learning network. Besides, give your method an impressive name may also help readers understand and remember your contributions in this article.
Comments on the Quality of English Language
1, Figure 2 is the architecture of your proposed method. A more thorough explanation especially about the novelty and details of your method could help readers understand your contributions in a short time. It could also be moved forward to the position of Figure 1 echoing previous contributions in line 75, page 2.
2, In line 35, page 2, ToF could not be “degraded”. I guess the author want to express this sentence as “which degrades the accuracy of the Time of Flight (ToF) based range estimation method”.
3, In line 75, page 2, the contributions of this article could be listed as several items marked 1, 2, 3… Besides, “the corrected the range estimations” should be “corrected range estimations”.
4, In line 137, page 4, “happens in two phases” could be substituted by “is composed of two part”. The same as line 170.
5, The written expression of this article may not be so fluent in English. Maybe a native English editor could help polish your article.
Reviewer 2 Report
Comments and Suggestions for Authors
The methodology is well-documented, the experimental evaluation is rigorous, and the results demonstrate the effectiveness and robustness of the proposed method. An analysis of the algorithm's time complexity and computational demands, along with a comparison to other existing localization methods, would provide a clearer understanding of its feasibility for deployment in resource-constrained devices or real-time applications.
Reviewer 3 Report
Comments and Suggestions for Authors
Comments to the Author
This paper proposes a novel localisation scheme using location-aware ranging correction on ultra wideband signals. However, there are several points that need to be addressed to improve the quality of the manuscript.
Suggestions to improve the quality of the paper are provided below:
1. In the introduction section, the authors should mention about a few application areas and location-based services that requires accurate location awareness to cater to a general audience. Please kindly review the following works as a good starting point and include other relevant application areas as you see fit.
Location-based building emergency response
10.1109/IUCC-CSS.2016.013
Location-based smart energy management
https://doi.org/10.1016/j.buildenv.2022.109472
Location-based smart HVAC controls
https://doi.org/10.1145/2517351.2517370
Location-based point-of-interest identification
https://doi.org/10.3390/ijgi10110779
Location-based occupancy prediction
https://doi.org/10.1016/j.buildenv.2022.109689
2. In the second paragraph of the Introduction section, the authors went directly into the advantages of UWB without providing a proper comparison with the other radio frequency-based technologies. Wifi and Bluetooth low energy technologies are two of the most popular approaches for indoor localisation due to their high positioning accuracy and low setup cost. Please kindly review the following work as a starting point and provide a comparison between these technologies.
https://doi.org/10.1016/j.buildenv.2020.106681
3. I strongly suggest that the authors take some time to list out the contributions of this work and how it extends upon the existing studies instead of simply providing a summary of what is done in this paper.
4. In the Results, the authors highlighted that there were signs of overfitting for Method (b) based on the results reflected in Table 2. However, it is unclear why Method (b) was prone to overfitting. This needs to be explained in more details. Additionally, Method (c) seem to be resistant to overfitting even though it also uses channel information features just like Method (b). Why is that the case? Please elaborate in the relevant subsections in Section 5.
5. Please include a Discussion section and discuss about the limitations of this work and how they will be addressed in future works.
Comments on the Quality of English Language
There are no major issues related to the manuscript's quality of English, except for some minor issues highlighted in my current set of comments.
Round 2
Reviewer 3 Report
Comments and Suggestions for Authors
Thank you for taking the time to address my comments thoroughly and comprehensively. I believe all my comments have been adequately addressed, and the quality of the manuscript has increased significantly as a result. I have determined that the manuscript is now ready for publication.
Comments on the Quality of English Language
There are no major issues related to the manuscript's quality of English, except for some minor issues that do not affect the clarity and flow of the manuscript.